# Variations in NSP1 of Porcine Reproductive and Respiratory Syndrome Virus Isolated in China from 1996 to 2022

**DOI:** 10.3390/genes14071435

**Published:** 2023-07-12

**Authors:** Zhiqing Zhang, Hang Zhang, Qin Luo, Yajie Zheng, Weili Kong, Liangzong Huang, Mengmeng Zhao

**Affiliations:** 1School of Life Science and Engineering, Foshan University, Foshan 528000, China; zhangzhiqing2023@outlook.com (Z.Z.); luoqin121104@163.com (Q.L.); zhengyajie2022@163.com (Y.Z.); 2Gladstone Institutes of Virology and Immunology, University of California, San Francisco, CA 94158, USA; weili.kong@gladstone.ucsf.edu

**Keywords:** porcine reproductive and respiratory syndrome virus, porcine reproductive and respiratory syndrome, NSP1, variations

## Abstract

Since its successful isolation in China in 1995, the porcine reproductive and respiratory syndrome virus (PRRSV) has been mutating into highly pathogenic strains by constantly changing pathogenicity and genetic makeup. In this study, we investigated the prevalence and genetic variation of nonstructural protein 1 (NSP1) in PRRSV-2, the main strain prevalent in China. After formulating hypotheses regarding the biology of the NSP1 protein, the nucleotide and amino acid similarity of NSP1 were analyzed and compared in 193 PRRSV-2 strains. The results showed that NSP1 has a stable hydrophobic protein with a molecular weight of 43,060.76 Da. Although NSP1 lacked signal peptides, it could regulate host cell signaling. Furthermore, NSP1 of different strains had high nucleotide (79.6–100%) and amino acid similarity (78.6–100%). In the amino acid sequence comparison of 15 representative strains of PRRSV-2, multiple amino acid substitution sites were found in NSP1. Phylogenetic tree analysis showed that lineages 1 and 8 had different evolutionary branches with long genetic distances. This study lays the foundation for an in-depth understanding of the nature and genetic variation of NSP1 and the development of a safe and effective vaccine in the future.

## 1. Introduction

Porcine reproductive and respiratory syndrome (PRRS), an acute and highly contagious infectious disease caused by the PRRS virus (PRRSV), is characterized by adverse reproductive outcomes and respiratory symptoms in pigs [1]. Regardless of age, most PRRSV-infected pigs experience breathing difficulties [2]. When infected, pregnant sows suffer from miscarriages, premature births, stillbirths, and mummified fetuses. Boars experience shortness of breath and reduced semen quality [3]. Infected piglets become anorexic, breathless, and prone to coughing [4]. Sick pigs infected with PRRS often die from secondary infections, such as pleurisy, streptococcosis, and gasping disease [5]. PRRS has inflicted heavy economic losses on pig farms [6] and gravely threatened the development of Chinese pig industry.

PRRSV is a single-stranded positive-sense enveloped RNA virus. The diameter of the viral particles is between 50 and 65 nm, and the surface has apparent spikes [5]. The nucleocapsid is a three-dimensional symmetrical icosahedral with a diameter between 25 and 35 nm [7,8]. The PRRSV genome is approximately 15 kb long and contains eight open reading frames (ORFs) [9,10]. The replicase gene ORF1 is the largest ORF at approximately 12 kb, which accounts for 80% of the viral genome. The position of ORF1 is after the 5′ noncoding region sequence. ORF1 includes ORF1a and ORF1b, and ORF1a encodes the nonstructural proteins NSP1–NSP8 [11,12,13,14]. PRRSV is divided into two genotypes, PRRSV-1 and PRRSV-2. Their nucleotide similarity is 55–70%, and amino acid similarity is 50–80% [15]. PRRSV-2 is the predominant strain in China. After analyzing more than 8500 PRRSV-2 ORF5 sequences, a global PRRSV-2 epidemic strain classification system was used to divide the global epidemic strains into nine lineages and thirty-seven sublineages [16].

The PRRSV-2 strain prevalent in China belongs to four lineages, lineage 1, 3, 5, and 8. Lineage 1 is a new epidemic lineage in China [17]. Its strains are collectively called NADC30-like strains, which undergo genetic recombination with domestic epidemic strains. Its representative strains are HNyc150, Chsx1401, JL580, FJ1402, FJW05, and TJnh1501 [15,18,19]. Lineage 3 is a late-onset strain, representing GM2 and QYYZ [20]. Lineage 5 is a long-running strain in China with a low clinical detection rate. Its representative strain BJ-4 has a high similarity with VR2332 [20]. Lineage 8 is common in China and is further divided into subgroups I–V. Subgroup I most resembles the genetic information of early isolated strains. The representative strains of subgroup I are CH-1a, BJ0706, NB-04, and JXA1, of which CH-1a is the earliest strain isolated in China [16]. The modified live virus (MLV)vaccine CH-1R was developed on the basis of the CH-1a [21]; it is widely used in China and provides over 70% cross protection against (HP-PRRS) HuN-4 strain [22] and provides 40% cross protection against (HP-PRRSV) TJ-F5 strain [23]. It induces lower levels of immune response and viremia [24]. Previous experiments have demonstrated the existence of genetic recombination between lineage 1, 3, 5, and 8 [25,26,27].

NSP1 is an inducing protein in PRRSV that influences host cells by regulating their immune response, blocking their tolerance to inflammation, and inhibiting signaling in specific cases. Moreover, NSP1 possesses hydrolase activity [28]. PRRSV mainly infects porcine alveolar macrophages, and NSP1 expression enhances the host’s inflammatory response by recruiting other immune cells around the infected macrophages [29].

NSP1 has two subunits, the conserved NSP1α and the variable NSP1β. At 18 °C, NSP1 is cleaved into NSP1α and NSP1β. At 37 °C, NSP1 is recombined into full-length NSP1 protein [30]. NSP1α protein contains three domains: (1) Met1–Glu65, which is the N-terminal zinc finger structure; (2) Pro66–Gln166, which is the papain-like protease domain; and (3) Arg167–Met180, which is the C-terminal extension region [31,32]. NSP1α is a multifunctional nuclear protein that regulates host interferon (IFN) [33]. Its N-terminal zinc finger structure has four cysteine binding sites with zinc ions, namely, Cys8, Cys10, Cys25, and Cys28. NSP1α inhibits IFN-β transcription at the N-terminal zinc finger structure and 167–177 amino acid (aa) regions by de-grading the CREB-binding protein and inhibiting the formation of enhancers [34,35,36]. The N-terminal zinc finger structure can also inhibit the NOD-like receptor family pyrin domain-containing 3 (NLRP3) inflammasome activity in the host system [37]. There are 14 amino acids at the end of the C-terminal extension of NSP1α that play an important role in maintaining the immunosuppressive activity of NSP1α against IFN-β and nuclear transcription factor-κB (NF-κB) [33,38].

Furthermore, NSP1β suppresses the host’s innate immunity and inhibits IFN synthesis and signaling. It has four domains: (1) Ala1–Ser48, which is the N-terminal domain (NTD); (2) Phe49–Thr84, which is the articulation region of the N- and C-terminals; (3) Val85–Pro181, which is the papain-like cysteine auto protease domain β (PCPβ) of papain-like cysteine protease structural region; and (4) Asn182–Gly203, which contains constitutive transport elements (CTEs) [39]. When PCPβ is blocked, it affects the gene replication of PRRSV [40]. PCPβ cleaves NSP1/NSP2 sites and releases NSP1β from downstream NSP2 [41]. Cys276 and His345 are key amino acid sites of NSP1β [40]. NSP1β also inhibits the expression, synthesis, and signal transduction of IFN [42]. The NTD of NSP1β participates in the ubiquitin protein degradation of karyopherin alpha 1 (KPNA1), which increases the ubiquitination and shortens the half-life of KPNA1, thereby blocking the nuclear translocation of the interferon-stimulated gene factor 3 complex [43,44].

Because NSP1 is involved in inhibiting host cell immune regulation, promoting PRRSV replication and helping PRRSV evade the intrinsic antiviral immunity of the host [45], we used NSP1 as the target gene for the genetic evolution analysis of PRRSV. Additionally, NSP1α has been found to suppress the production of IFN-β, while NSP1β hinders both the synthesis and signaling of IFN, effectively blocking the production of IFN-I in the body. Zhao’s experiments have demonstrated that the NSP1β mutant vaccine strain can effectively combat the reversion to virulence that often occurs after vaccination, while also stimulating a specific natural immune response [46]. This suggests that further investigation into genetic mutations in NSP1 could potentially lead to the development of vaccines that offer improved cross protection and higher immune efficacy.

In this study, the genetic variation of PRRSV NSP1 was investigated by analyzing the nucleotide and amino acid sequences of NSP1 of 193 PRRSV-2 strains to construct a phylogenetic tree, and the amino acid sequences of NSP1 were compared to reflect the amino acid substitution of NSP1 in different lineages. The findings of this study provide a basis for further screening of amino acid loci affecting PRRSV virulence and for further studies on the regulatory role of NSP1 amino acid locus changes on the host cellular immune response and the impact on PRRS pathogenesis. We also discussed the role of NSP1 in regulating the immune response of host cells. Hence, our study on NSP1 mutations provides a theoretical basis for preparing a PRRSV vaccine to facilitate the prevention and control of PRRS. The approval number of the Foshan University Ethical committee is SYXK-2020-0235.

## 2. Materials and Methods

### 2.1. Source of Sequences

The nucleotide information of NSP1 of the 193 PRRSV strains used in the experiments was obtained from GenBank, and the amino acid sequences of the virulent strains were translated on EditSeq function of DNASTAR software (version 7.0, Madison, WI, USA).

### 2.2. Sequence Analysis of PRRSV NSP1

For the CH-1R strain, the main physicochemical properties of NSP1 were analyzed using the ProtParam tool of the ExPASy resource portal (https://web.expasy.org/protparam, accessed on 27 August 2022) [47]. Signal peptides were predicted using SignalP-6.0 (https://services.healthtech.dtu.dk/service.php?SignalP-6.0, accessed on 27 August 2022) [48]. The information about reference strains is shown in Table 1.

### 2.3. Nucleotide Sequence Comparison in NSP1

The MegAlign function of DNASTAR software (version 7.0, Madison, WI, USA) was used to analyze the nucleotide sequence similarity in NSP1. The information about reference strains is shown in Table 1.

### 2.4. Amino Acid Sequence Comparison in NSP1

The similarity of NSP1 amino acid sequences was analyzed using the Clustal W method in the MegAlign function of DNASTAR software (version 7.0, Madison, WI, USA). The amino acid sequences of representative strains of each lineage were compared using MEGA software (version 7.0; Center for Evolutionary Medicine and Informatics, Tempe, AZ, USA). GeneDoc [49] annotated the amino acid sequence comparison chart. The information about reference strains is shown in Table 1.

### 2.5. Phylogenetic Tree Analysis

The neighbor-joining tree (NJ) and the maximum likelihood tree (ML) of 193 strains of PRRSV based on NSP1 were generated using MEGA software (version 7.0; Center for Evolutionary Medicine and Informatics, Tempe, AZ, USA). In testing the neighbor-joining tree, the phylogeny test w MEGA software (version 7.0; Center for Evolutionary Medicine and Informatics, Tempe, AZ, USA) [50] as the bootstrap method, the number of bootstrap replications was 1000, and the p-distance was utilized. In testing the maximum likelihood tree, the model was the Tamura–Nei model. iTOL (iTOL: Interactive Tree Of Life (embl.de)) [51] annotated the phylogenetic tree. The information about 192 strains is shown in Table 1.

## 3. Results

### 3.1. Sequence Analysis Equations

#### 3.1.1. Base Distribution and Physicochemical Properties of NSP1

The total length of the PRRSV-2 NSP1 sequence was 1149 bp, corresponding to 383 amino acids. Among these, 242 were adenine (A) (21.06% of total bases), 308 were guanine (G) (26.81% of total bases), 280 were thymine (T) (24.37% of total bases), and 319 were cytosine (C) (27.76% of total bases). Furthermore, 522 were AT (45.43% of total bases), and 627 were CG (54.57%). Using the EditSeq function in the DNASTAR software package, the NSP1 encoded 42 highly basic amino acids (K and R), 36 highly acidic amino acids (D and E), 141 hydrophobic amino acids (A, I, L, F, W, and V), and 84 hydrophilic amino acids (N, C, Q, S, T, and Y).

#### 3.1.2. Stability Coefficient and Signal Peptide Prediction of NSP1

ExPASy analysis showed that the predicted molecular weight of NSP1 was 43,060.76 Da, and the instability coefficient was 52.76, implying that it is an unstable protein. The fat solubility coefficient was 83.99, and the theoretical isoelectric point was 8.71. The total mean of hydrophilicity was −0.145, indicating that NSP1 is a hydrophobic protein. SignalP-6.0 estimated that the protein encoded by NSP1 had no signaling peptides.

### 3.2. Nucleotide Similarity of NSP1

The nucleotide similarity of NSP1 of 193 PRRSV-2 strains was compared and analyzed. The results showed that the nucleotide similarity was 79.6–100%. The lowest similarity level was for the SCcd2020 and YL strains and the SCya18 and YC strains (79.6%), whereas the highest was for the GX1002, JXA1-P45, JXA1-P70, JXA1-P80, and rJXA1-R strains (100%). Furthermore, the nucleotide sequences were identical for JXA1P10 and JXA1P15, rV63 and rV68, QY2010 and QYYZ, XJU-1 and JSTZ1907-714, and PRRSV03 and WUH1 strains.

Because the nucleotide similarity analysis plot of 193 strains of NSP1 was too large, we selected representative strains of each lineage for nucleotide analysis to obtain a comparative plot and elucidate the genetic variation of NSP1 in the evolution of PRRSV. As illustrated in Figure 1, the nucleotide similarity was 89.5–93.6% in the lineage 1 group, 85.8% in the lineage 3 group, 99.7% in the lineage 5 group, 93–99.9% in the lineage 8 group, 98.8–99.9% in the C-PRRSV-like group, and 99.3–99.7% in the HP-PRRSV-like group. Among these, the nucleotide sequence of the lineage 3 group showed the most considerable difference (Figure 1).

### 3.3. Amino Acid Sequence Similarity of NSP1

The results showed that the amino acid similarity of NSP1 of 193 PRRSV-2 strains was 78.6–100%. The lowest similarity was observed for the FJM4 and CH2002 strains (78.6%). The highest similarity was observed for the BJBLZ, FZ06A, HUN4, SHH, and TJ strains and the BJSY-1, GD, HUB1, HUB2, and Jiangxi-3 strains (100%). Furthermore, the amino acid sequences were the same for CH-1a and CH-1R; CH2003, CH2004, and CHsx1401; 07QN and JXA1; JSTZ1907-714 and XJu-1; PRRSV02, PRRSV03, and TP; and BJ-4 and rV63 strains.

Because the amino acid sequence similarity plot of 193 strains of NSP1 was too large, we selected representative strains of each spectrum for nucleotide sequence analysis to obtain a comparative plot and elucidate the genetic variation of NSP1 in the evolution of PRRSV. As shown in Figure 2, the amino acid similarity was 89.3–91.6% in the lineage 1 group, 87.5% in the lineage 3 group, 99.5% in the lineage 5 group, 90.1–100% in the lineage 8 group, 97.7–100% in the C-PRRSV-like group, and 99.7–100% in the HP-PRRSV-like group. Among these, the amino acid sequence in the lineage 3 group showed the most significant difference (Figure 2).

### 3.4. Amino Acid Sequence Alignment of NSP1 of Representative Strains

The amino acid sequences of the representative strains of each lineage were compared using MEGA software (Figure 3). The amino acid sequence of NSP1 PRRSV-2 consisted of 383 amino acid residues, with some substitutions and no additions or deletions. The NSP1 amino acid sequence was generally relatively conservative, particularly the 1–97 sites with low substitution frequency. However, among the 182–291 and 302–378 sites, the sequences of various pedigrees were different, and some substitutions were observed.

At the sites 183, 196, 198, 226, 234, 321, 349, 368, and 370, after comparison with the VR2332 and BJ-4 strains of lineage 5, the strains of lineages 1, 3, and 8 had amino acid substitutions, whereas the VR2332 and BJ-4 strains of lineage 5 did not. At the sites 202, 208, 209, 219, 223, 236, 249, 257, 263, 267, 276, 285, 291, and 319, the lineage 1 strains had amino acid substitutions. At sites 227, 232, 289, 368, and 371, three C-PRRSV-like strains had amino acid substitutions.

Finally, the amino acid substitution site of the new wild strain GDZQ-2021 was consistent with the traditional representative strains of HP-PRRSV-like and C-PRRSV-like groups.

### 3.5. Phylogenetic Tree Analysis

Phylogenetic tree analysis showed that the QYYZ and QY2010 strains of lineage 3 had close genetic distance with SDRZ01, HH08, CH2002, and CH-1a strains of lineage 8.1, whereas the VR2332 strain of lineage 5 had a long genetic distance with HNyc15, CHsx1401, and BJ-F1501 strains of lineage 8.7 (Figure 4 and Figure 5).

## 4. Discussion

In recent years, PPRSV has been consistently evolving and enhancing virulence, owing to genetic recombination and variation [52]. PRRS was first identified in the United States and Europe in 1986 and 1990 [53]. PRRSVs with antigenicity close to the United States-based strain were isolated in Japan in 1990 [54]. PRRSV appeared in China in 1995. In 1996, the virus was isolated by Guo et al. and named CH-1a [55]. In 1997, Yang et al. isolated BJ-4 [56]; CH-1a and BJ-4 are North-America-based strains. After spreading in China, the genetic diversity of PRRSV gradually increased, and the amino acids of NSP2 and ORF5 are highly pathogenic [57,58]. After the PRRSV outbreak in China, Tian et al. compared genomics through a trial. The analysis determined that the strain was of a North American genotype with high nucleotide sequence similarities with the Chinese HB-1 strain. Moreover, the NSP2 of the strain was analyzed using bioinformatics analysis. The results confirmed a discontinuous absence of 30 amino acids, which may have contributed to its high pathogenicity [59]. In 2014, Liu et al. compared 36 isolated PRRSV strains and glycoprotein 5 of eight vaccine strains. Similarity comparison and amino acid analysis revealed that the 2014 isolates had variations in the neutralizing expression of independent and N-glycosylation sites compared with the first subset of strains [60]. In 2017, Long confirmed through amino acid sequence analysis of NSP2 that the NADC34-like strain was introduced to China from North America. This strain and China’s native strain, HP-PRRSV, eventually recombined to form a new strain, SCcd2020 [61]. Therefore, PRRSV diversity is ever increasing.

Although the changes in NSP1 were not as evident as those in the highly variable NSP2 and ORF5, NSP1 still underwent considerable variation that impacted PRRSV evolution. Our analyses showed that NSP1 is a hydrophobic protein without any signaling peptides, indicating that NSP1 is not transportable and may be synthesized in the cytoplasm or organelle matrix from proteins composed of free ribosomes that can enter the cytosol and not from secreted or membrane proteins. NSP1 has high hydrophobicity, which is conducive to forming an α-helix and the inward folding of protein to form a secondary structure; therefore, it is considered stable. Nucleotide and amino acid similarity analyses of 193 PRRSV-2 NSP1 proteins showed that the nucleotide similarity was 79.6–100%, and the amino acid similarity was 78.6–100%. As shown in Figure 1, the lineage 3 strain FJFS had high similarity with the lineage 8 strain TJ, which was as high as 94.2%. This finding indicates that NSP1 undergoes genetic variation during evolution; moreover, there may be gene recombination between the two lineages, which needs to be further verified by recombinant analysis of the strains of the two lineages. Furthermore, lineage 3 is a late-onset strain, and lineage 8 is common in China, meaning that their survival rates differ despite their high nucleotide similarity in NSP1. This result implies that the effect of NSP1 may not be the main factor affecting the survival of PRRSV in the host, which warrants further study. As illustrated in Figure 2, the amino acid sequence of NSP1 had a high similarity. Regarding the differences within the lineage amino acid sequences, lineages 3 and 8 had the largest and smallest differences, respectively. The similarity within a pedigree was higher than that between pedigrees.

Based on the above results, we conclude that during the evolution of PRRSV, NSP1 is relatively conserved. However, genetic variation still occurs, and there may be recombination between strains of different pedigrees, which is worthy of further experimental confirmation. In the comparative analysis of amino acid sequences, NSP1 amino acids underwent substitutions at individual sites, and no additions or deletions were observed (Figure 3). The variation in NSP1 was more conserved than that of NSP2 and ORF5. In previous studies, Shi found that the smallest region of the carboxyl-terminal extension required for NSP1α to inhibit IFN-β transcription was the 167–176 amino acid site, in which the 176th amino acid and zinc finger structure played the most important roles [34,62]. Replacing the 176th amino acid significantly impacted the inhibition of IFN-β transcription. Shi et al. also found that Cys-270 and His-339 were the catalytic residues of NSP1β; therefore, if Cys were mutated to Ser or His to Ala, NSP1β would lose its enzymatic activity [38]. Wang et al. revealed that KPNA1 degradation and IFN-mediated signal inhibition decreased when residue 19 of NSP1β changed from Val to Ile [44]. These studies indicate that amino acid substitution is related to the viral inhibition of IFN transcription, which impacts virus virulence, phagocytosis, and cleavage.

The phylogenetic tree analyses in Figure 4 and Figure 5 show a relatively large genetic distance between lineages 1 and 8. Zhou et al. proved a genetic recombination between these two strains [62], which increased the survival rate of PRRSV in the host. Therefore, these two lineages are common in China. In genetics, species are determined by conserved, irreplaceable sequences that ensure species stability. In contrast, the differences within species are caused by non-conservative sequences. The recombination and variations in genetic information leads to new strains with different viability and virulence properties [63].

As NSP1 is a conserved sequence, its characteristics enable PRRSV to influence host cells by effectively regulating their immune response, inhibiting signal transduction in specific cases, and blocking their tolerance to inflammation. As the main protease of PRRSV, NSP1 can be used to study corresponding drugs according to their characteristics. In previous studies, Xue et al. studied the three-dimensional structure of PRRSV NSP1β and revealed that the CTD possessed a cleavage site of Trp-Tyr-Gly203 ↓ Ala-Gly-Lys. In contrast, the NTD had strict metal-dependent nuclease activity. In addition, their mutagenesis studies confirmed that Lys18 and Glu52 in the NTD surface-charged region contributed to NSP1β nuclease activity [64]. These results can aid in developing more multi-target drugs in the future. Shi et al. found that HnRNP A2/B1 overexpression can enhance PRRSV virulence without affecting the IFN-β transcription signaling pathway, implying a different mechanism is involved [34]. These studies provide a theoretical basis for future vaccine development. The findings of the present study on the genetic variation of NSP1 pave the way for further research on its role in vaccine development and the pathogenesis, prevention, and control of PRRSV.

## 5. Conclusions

The data showed that the strains of the same lineage and different lineages had high nucleotide and amino acid similarity regarding NSP1. The amino acid sequence is relatively conserved, and there are amino acid substitutions at some sites. The study discusses the role of NSP1 in the PRRS and contributes to the monitoring of the evolution of PRRSV in China and the development of methods to prevent and control PRRS. The genetic variation analysis of NSP1 in this study may provide a theoretical basis for the potential impact of NSP1 genetic variation on PRRSV virulence and merits further study in the future. It also provides a basis for future research into more cross-protective vaccines.

## Figures and Tables

**Figure 1 genes-14-01435-f001:**
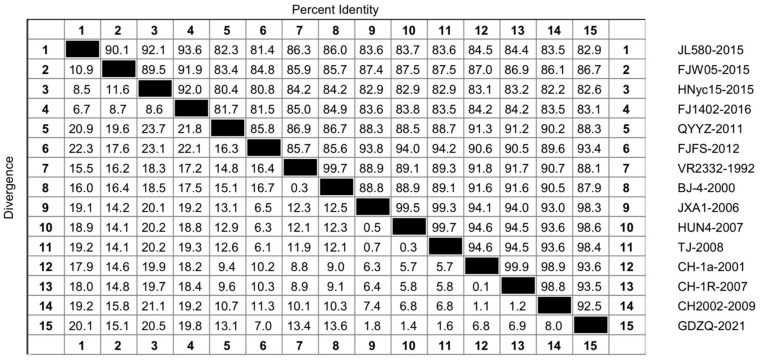
Nucleotide sequence similarity analysis of NSP1 of 15 representative lineage strains 1, 3, 5, and 8. The strains were aligned by the Clustal W method and viewed by sequence distance in the MegAlign function of DNASTAR software. The strains included JL580, FJW05, HNyc15, and FJ1402 in lineage 1; QYYZ and FJFS in lineage 3; VR2332 and BJ-4 in lineage 5; JXA1, HUN4, and TJ in the HP-PRRSV-like group of lineage 8; and CH-1a, CH-1R, and CH2002 in the C-PRRSV-like group of lineage 8. Furthermore, the GDZQ strain was a new wild strain in lineage 8.

**Figure 2 genes-14-01435-f002:**
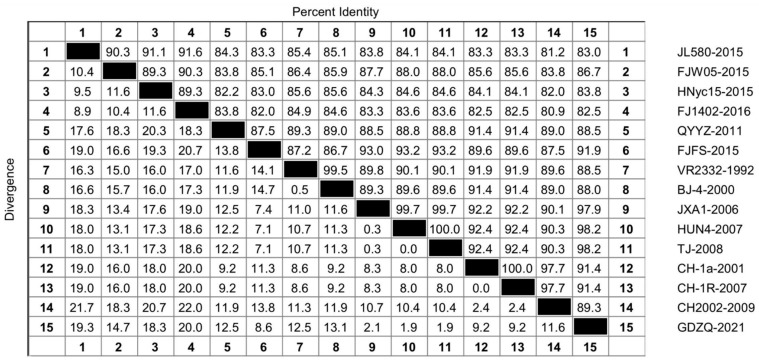
Amino acid sequence similarity analysis of NSP1 of 15 representative lineage strains 1, 3, 5, and 8. The strains were aligned by the Clustal W method and viewed by sequence distance in the MegAlign function of DNASTAR software. The strains included JL580, FJW05, HNyc15, and FJ1402 in lineage 1; QYYZ and FJFS in lineage 3; VR2332 and BJ-4 in lineage 5; JXA1, HUN4, and TJ in the HP-PRRSV-like group of lineage 8; and CH-1a, CH-1R, and CH2002 in the C-PRRSV-like group of lineage 8. Furthermore, the GDZQ strain was a new wild strain in lineage 8.

**Figure 3 genes-14-01435-f003:**
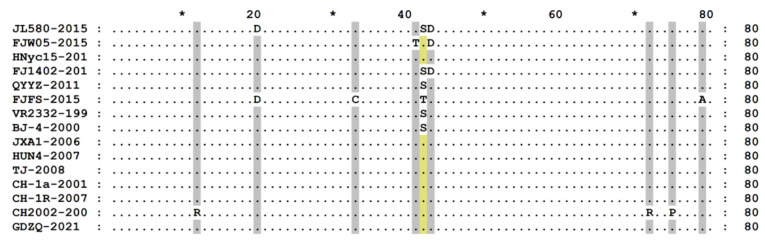
Amino acid sequence alignment of 15 representative lineage strains 1, 3, 5, and 8. The amino acid sequences of the representative strains of each lineage were compared using MEGA software. The strains included JL580, FJW05, HNyc15, and FJ1402 in lineage 1; QYYZ and FJFS in lineage 3; VR2332 and BJ-4 in lineage 5; JXA1, HUN4, and TJ in the HP-PRRSV-like group of lineage 8; and CH-1a, CH-1R, and CH2002 in the C-PRRSV-like group of lineage 8. Further, the GDZQ strain was a new wild strain in lineage 8. * means Conservative amino acid.

**Figure 4 genes-14-01435-f004:**
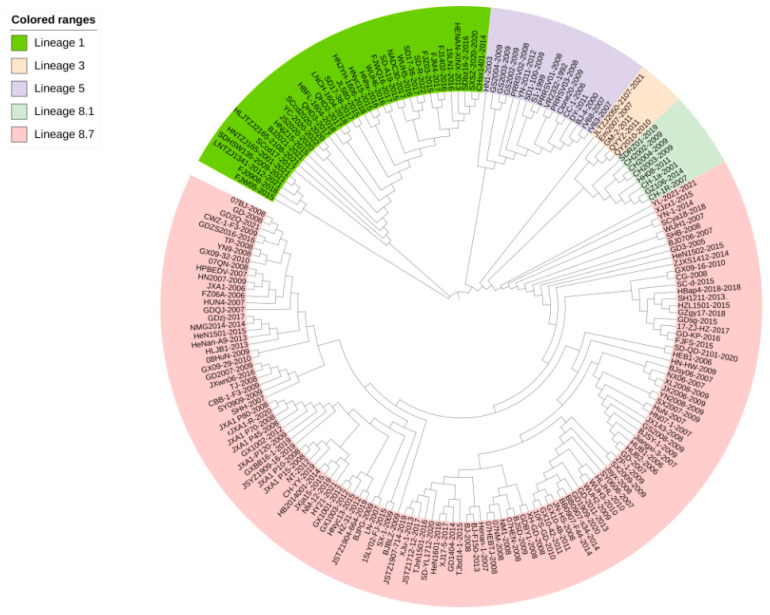
The phylogenetic tree constructed by the neighbor-joining method (NJ) on NSP1 sequences of 193 PRRSV strains. Phylogenetic tree analysis of NSP1 gene constructed using MEGA software (version 7.0; Center for Evolutionary Medicine and Informatics, Tempe, AZ, USA). The neighbor-joining method is supported using the p-distance model with uniform rates.

**Figure 5 genes-14-01435-f005:**
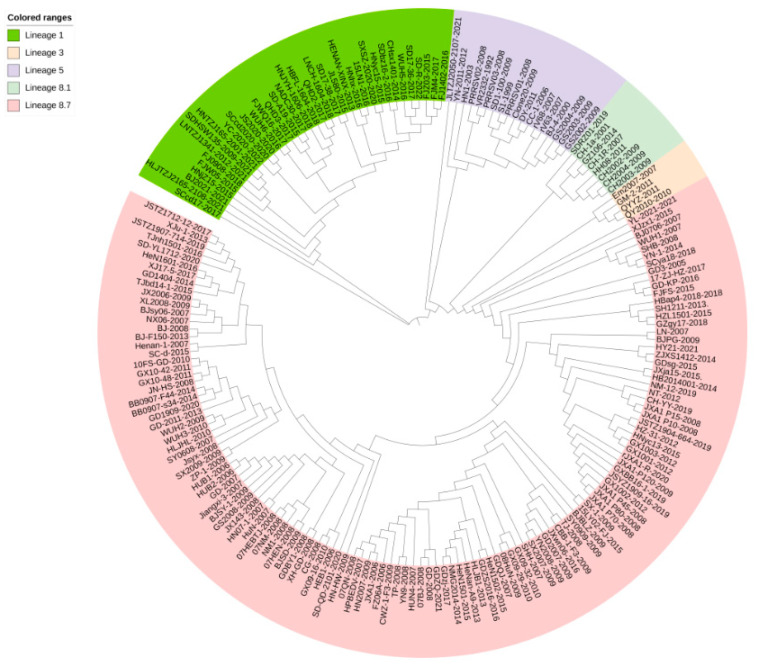
The phylogenetic tree constructed by the maximum likelihood method (ML) on NSP1 sequences of 193 PRRSV strains. Phylogenetic tree analysis of NSP1 gene constructed using MEGA software (version 7.0; Center for Evolutionary Medicine and Informatics, Tempe, AZ, USA). The maximum likelihood method is supported using the Tamura–Nei model with uniform rates.

**Table 1 genes-14-01435-t001:** Reference information for each PRRSV strain.

Year	Area	Strain	GenBank
			Accession Number
1992	USA	VR2332	EF536003
1999	USA	S1	MW592733
2000	China	BJ-4	AF331831.1
2001	China	CH-1a	AY032626.1
2003	China	HN1	AY457635.1
2006	China	JXA1	EF112445
2006	China	HUB2	EF112446
2006	China	HUB1	EF075945.1
2006	China	HEB1	EF112447.1
2006	China	FZ06A	MF370557.1
2006	China	CC-1	EF153486.1
2007	China	WUH1	EU187484
2007	China	SHH	EU106888
2007	China	rV63	EU360129
2007	China	rV68	EU360128
2007	China	SY0608	EU144079
2007	China	NX06	EU097706
2007	China	LN	EU109502
2007	China	Jiangxi-3	EU200961
2007	China	HUN4	EF635006
2007	China	HuN	EF517962
2007	China	HPBEDV	EU236259.1
2007	China	HN07-1	KX766378.1
2007	China	Henan-1	EU200962.1
2007	China	GDQJ	GQ374441.1
2007	China	GD-2007	EU880433.2
2007	China	Em2007	EU262603.1
2007	China	CH-1R	EU807840.1
2007	China	BJsy06	EU097707.1
2007	China	BJ0706	GQ351601.1
2008	China	TP	EU864233
2008	China	YN9	GU232738
2008	China	TJ	EU860248
2008	China	SHB	EU864232
2008	China	XH-GD	EU624117
2008	China	PRRSV01	FJ175687
2008	China	PRRSV02	FJ175688
2008	China	PRRSV03	FJ175689
2008	China	NM1	EU860249
2008	China	JXA1_P80	FJ548853
2008	China	JXA1_P70	FJ548852
2008	China	JXA1_P45	FJ548851
2008	China	JXA1_P15	FJ548855
2008	China	JXA1_P10	FJ548854
2008	China	JX143	EU708726
2008	China	Jsyx	EU939312
2008	China	JN-HS	HM016158
2008	China	GDBY1	GQ374442.1
2008	China	GD	EU825724.1
2008	China	CG	EU864231.1
2008	China	BJ	EU825723.1
2008	China	07QN	FJ394029.1
2008	China	07NM	FJ393456.1
2008	China	07HEN	FJ393457.1
2008	China	07HEBTJ	FJ393458.1
2008	China	07BJ	FJ393459.1
2009	China	SY0909	HQ315837
2009	China	YN2008	EU880435
2009	China	XL2008	EU880436
2009	China	WUH2	EU678352
2009	China	SX2009	FJ895329
2009	China	SX2007	EU880434
2009	China	SD1-100	GQ914997
2009	China	SX-1	GQ857656
2009	China	ZP-1	HM016159
2009	China	JXA1-P120	KC422727
2009	China	JX2006	EU880432
2009	China	HN-HW	FJ797690.1
2009	China	HN2007	EU880437.2
2009	China	GS2008	EU880431.2
2009	China	GS2004	EU880443.3
2009	China	GS2003	EU880442.2
2009	China	GS2002	EU880441.2
2009	China	GD2007	EU880433.2
2009	China	CWZ-1-F3	FJ889130.1
2009	China	Clone20	FJ899592.1
2009	China	CH2004	EU880439.2
2009	China	CH2003	EU880440.2
2009	China	CH2002	EU880438.2
2009	China	CBB-1-F3	FJ889129.1
2009	China	BJSY-1	FJ950744.1
2009	China	BJSD	FJ950747.1
2009	China	BJPG	FJ950746.1
2009	China	BJBLZ	FJ950745.1
2009	China	08HuN	GU169411.1
2010	China	QY2010	JQ743666
2010	China	WUH3	HM853673
2010	China	HLJHL	HM189676.1
2010	China	GX09-32	HM214915.1
2010	China	GX09-29	HM214914.1
2010	China	GX09-16	HM214913.1
2010	China	10FS-GD	JX192634.1
2011	China	QYYZ	JQ308798
2011	China	HH08	JX679179.1
2011	China	GX10-48	JQ309823.1
2011	China	GX10-42	JQ309822.1
2011	China	GM2	JN662424.1
2011	China	DY	JN864948.1
2012	China	YN-2011	JX857698
2012	China	NT	KP998420
2012	China	HZ-31	KC445138
2012	China	GX1003	JX912249.1
2012	China	GX1002	JQ955658.1
2012	China	GX1001	JQ955657.1
2013	China	SH1211	KF678434
2013	China	XJu-1	KF815525
2013	China	HLJB1	KT351740.1
2013	China	HENAN-XINX	KF611905.1
2013	China	HeNan-A9	KJ546412.1
2013	China	GD-2011	KC527830.1
2013	China	BJ-F150	KP890342.1
2014	China	ZJXS1412	MF669722
2014	China	YN-1	KJ747052
2014	China	NMG2014	KM000066
2014	China	HB2014001	KM261784.1
2014	China	GZ106	KJ541663.1
2014	China	GD1404	MF669720.1
2014	China	CHsx1401	KP861625.1
2014	China	BB0907-s34	KM453698.1
2014	China	BB0907-F44	KM453699.1
2015	China	SD-A19	MF375260
2015	China	XJzx1-2015	KX689233
2015	China	SC-d	MF375261
2015	China	TJbd14-1	KP742986
2015	China	JXja15	KR149645
2015	China	JL580	KR706343
2015	China	HZL1501	MF669721
2015	China	HNyc15	KT945018.1
2015	China	HNyc13	KT022072.1
2015	China	HNjZ15	KT945017.1
2015	China	HeN1502	MF766473.1
2015	China	HeN1501	MF766472.1
2015	China	GDsg	KX621003.1
2015	China	GD3	GU269541.1
2015	China	FJZ03	KP860909.1
2015	China	FJW05	KP860911.1
2015	China	FJFS	KP998476.1
2015	China	15LY02-FJ	KU215417.1
2016	China	WUH6	KU523367
2016	China	WUH5	KU523366
2016	China	SDbz16-2	MH588710
2016	China	TJnh1501	KX510269
2016	China	JXwn06	EF641008
2016	China	HNhx	KX766379.1
2016	China	HeN1601	MF766474.1
2016	China	GDZS2016	MH046843.1
2016	China	GD-KP	KU978619.1
2016	China	FJ1402	KX169191.1
2016	China	15LN1	KX815423.1
2017	China	SD17-36	MH121061
2017	China	SD17-38	MH068878
2017	China	SCcd17	MG914067
2017	China	XJ17-5	MK759853
2017	China	QHD2	MH167387
2017	China	QHD3	MH167388
2017	China	JSTZ1712-12	MK906026
2017	China	GDzj	MF772778.1
2017	China	FJWQ16	KX758249.1
2017	China	FJM4	KY412888.1
2017	China	17-ZJ-HZ	MF770574.1
2018	China	SCya18	MK144543
2018	China	LNCH-1604	MH651741
2018	China	HNJYH-1606	MH651740.1
2018	China	HBFL-1604	MH651739.1
2018	China	HBap4-2018	MZ579701.1
2018	China	GZgy17	MK144542.1
2018	China	FJ0908	MK202794.1
2019	China	SDRZ01	MZ322956
2019	China	NM-12	MN026347
2019	China	JSYZ1909-16	MT780871
2019	China	JSTZ1907-714	MN547967
2019	China	JSTZ1904-664	MN547966
2019	China	GXBB16-1	MN026346.1
2019	China	CH-YY	MK450365.1
2020	China	YC	HM027595
2020	China	SD-YL1712	MT708500
2020	China	SD-QD-2101	MZ172971
2020	China	SCcd2020	MW803134
2020	China	SXSZ-2020	MW880772
2020	China	rJXA1-R	MT163314
2020	China	JS2020	MZ342900
2020	China	GD1909	MT165636.1
2021	China	SDHSW135-2009	OL516361
2021	China	YL-2021	MZ169406
2021	China	LNTZJ1341-2012	OL516360
2021	China	JLTZJ2050-2107	OL516359
2021	China	HY21	OL687155
2021	China	HNTZJ165-2001	OL516358.1
2021	China	HLJTZJ2165	OL516356.1
2021	China	BJ2021	OK095299.1
2022	China	SD-R	ON254650

## Data Availability

Data sharing not applicable. No new data were created or analyzed in this study. Data sharing is not applicable to this article.

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
