# Peer review of "Variations in NSP1 of Porcine Reproductive and Respiratory Syndrome Virus Isolated in China from 1996 to 2022"

_genes, 2023, doi:10.3390/genes14071435_

Round 1

Reviewer 1 Report

Major comments

Introduction

-        You could add a paragraph on the importance of NSP1 based on the demands of PRRSV vaccine development in the future.

Materials and Methods

-        L109: you should report the approval number of this study by the Institutional Ethical committee.

-        L111-115: add appropriate references.

-        L128-135: add appropriate references.

Conclusion 

-        You should expand the length of this part, underlying the novelty of your results.

Author Response

Dear reviewer 1:

Thank you for constructive comments. We have made some changes to the manuscript. These changes will not influence the content and framework of the manuscript. And here we list the changes and marked in yellow in the revised manuscript. We have also responded to each of your comments and suggestions in a point-by-point manner. We appreciate for your warm work earnestly and hope that the correction will meet with approval.

  1. You could add a paragraph on the importance of NSP1 based on the demands of PRRSV vaccine development in the future.

Response: We have added the importance of NSP1 based on the demands of PRRSV vaccine development in the future in lines 107-113.

  1. Materials and Methods. L109: you should report the approval number of this study by the Institutional Ethical committee.

Response: We have added the approval number of the Foshan University Ethical committee, they are in lines 123-124.

  1.    L111-115: add appropriate references.

Response: We have added appropriate references in lines 133-134.

  1. L128-135: add appropriate references.

Response: We have added appropriate references in lines 150,152,155.

  1. Conclusion: You should expand the length of this part, underlying the novelty of your results.

Response: We have expanded the length of conclusions to reflect the usefulness and novelty of our results in lines 344-350.

Regards!

Mengmeng Zhao

Liangzong Huang

Reviewer 2 Report

Minor

At line 111 the CH-1R is not been described in full before in the text.

In Materials and methods it was not described what method you used to obtain the useful material for the sequencing analyses.

For the figure 1 and 2 it is possible to insert a line space?

For the Bibliography it is possible to insert more recent references?

Author Response

Dear reviewer 2:

Thank you for constructive comments. We have made some changes to the manuscript. These changes will not influence the content and framework of the manuscript. And here we list the changes and marked in yellow in the revised manuscript. We have also responded to each of your comments and suggestions in a point-by-point manner. We appreciate for your warm work earnestly and hope that the correction will meet with approval.

  1. At line 111 the CH-1R is not been described in full before in the text.

Response: We have added the descriptions of CH-1R in lines 64-70.

  1. In Materials and methods it was not described what method you used to obtain the useful material for the sequencing analyses.

Response: We have added the descriptions of what method you used to obtain the useful material for the sequencing analyses in lines 126-129.

  1. For the figure 1 and 2 it is possible to insert a line space?

Response: We have added a row of 187 after Figure 1, and a row of 210 after Figure 2.

  1. For the Bibliography it is possible to insert more recent references?

Response: We have quoted 13 new references for the years 2020 to 2023.

Regards!

Mengmeng Zhao

Liangzong Huang